# Linear Mode Connectivity between Multiple Models modulo Permutation Symmetries

**Akira Ito** [1]  **Masanori Yamada** [1]  **Atsutoshi Kumagai** [2 1]

## Abstract

Ainsworth et al. (2023) empirically demonstrated that linear mode connectivity (LMC) can be achieved between two independently trained neural networks (NNs) by applying an appropriate parameter permutation. LMC is satisfied if a linear path with non-increasing test loss exists between the models, suggesting that NNs trained with stochastic gradient descent (SGD) converge to a single approximately convex low-loss basin under permutation symmetries. However, Ainsworth et al. (2023) verified LMC for two models and provided only limited discussion on its extension to multiple models. In this paper, we conduct a more detailed empirical analysis. First, we show that existing permutation search methods designed for two models can fail to transfer multiple models into the same convex low-loss basin. Next, we propose a permutation search method using a straight-through estimator for multiple models (STE-MM). We then experimentally demonstrate that even when multiple models are given, the test loss of the merged model remains nearly the same as the losses of the original models when using STE-MM, and the loss barriers between all permuted model pairs are also small. Additionally, from the perspective of the trace of the Hessian matrix, we show that the loss sharpness around the merged model decreases as the number of models increases with STE-MM, indicating that LMC for multiple models is more likely to hold. The source code implementing our method is available at https://github.com/e5-a/STE-MM.

[1]NTT Social Informatics Laboratories [2]NTT Computer and Data Science Laboratories. Correspondence to: Akira Ito <akira.itoh@ntt.com>.

*Proceedings of the 42nd International Conference on Machine Learning*, Vancouver, Canada. PMLR 267, 2025. Copyright 2025 by the author(s).

## 1. Introduction

Deep neural networks have achieved great success in various fields, including image classification, speech recognition, and natural language processing (Vaswani et al., 2017; van den Oord et al., 2016; Zhao et al., 2023). The optimization of the large neural network (NN) models used in these applications poses a massive non-convex optimization problem. Nevertheless, stochastic gradient descent (SGD), despite its simplicity, consistently finds good solutions. One hypothesis for this seemingly contradictory situation is that the loss functions of NNs have a much simpler structure than we might imagine. Several prior studies have empirically shown that solutions of different NNs can be connected by simple low-loss nonlinear paths. Furthermore, in recent years, Entezari et al. (2022) hypothesized that the following conjecture holds when all permutation symmetries of NNs are taken into account.

**Conjecture 1.1** (Permutation invariance, informal). Let $\boldsymbol{\theta}_a$ and $\boldsymbol{\theta}_b$ be two SGD solutions (trained model parameters). With high probability, there exists a permutation $\pi$ such that the loss barrier between $\boldsymbol{\theta}_a$ and $\pi(\boldsymbol{\theta}_b)$ is sufficiently small.

Here, a barrier refers to the increase in the loss observed during linear interpolation between two models. When the barrier between two models is sufficiently small, we say that linear mode connectivity (LMC) holds between those models. Entezari et al. (2022) hypothesized that LMC can be achieved by considering the permutation symmetries of NNs. Subsequently, Ainsworth et al. (2023) and Singh & Jaggi (2020) empirically demonstrated that appropriate permutation search methods can identify permutations that satisfy Conjecture 1.1. Based on these findings, Ainsworth et al. (2023); Entezari et al. (2022) argued that the weights of trained multiple models can be transferred into a single approximately convex loss basin. If this claim holds, the optimization of NNs could be interpreted as approximately convex optimization, thereby explaining the effectiveness of SGD. In addition, analyzing LMC among multiple models under permutation symmetries is important not only for understanding how SGD works but also for model merging, where independently trained models are combined. For example, permutation-based model merging has the potential to combine multiple models without significant costs, such

as training. Indeed, several prior studies (Singh & Jaggi, 2020; Wang et al., 2020; Peña et al., 2023; Yamada et al., 2024) have proposed methods for model merging, federated learning, and continual learning based on the permutation symmetries of NNs.

In this paper, we investigate more deeply whether the weights of multiple models can be transferred into a single (approximately convex) loss basin. First, we show that gathering multiple models into a single loss basin is difficult by simply applying a permutation search method that targets only two models. We also demonstrate that, although some previous studies (Ainsworth et al., 2023; Crisostomi et al., 2024) have proposed permutation search methods for multiple models, the test loss and the accuracy of the merged model deteriorate as the number of models increases in these methods. To overcome this issue, we propose a novel permutation search method using a straight-through estimator for multiple models (STE-MM), and experimentally confirm its effectiveness.

**Contributions.** This paper makes the following three contributions:

- **Difficulty in satisfying LMC among three or more models.** We demonstrate that simply applying conventional permutation search methods, which focus only on two models, makes it difficult to establish LMC among multiple models. In addition, we show that even when using existing permutation search methods for multiple models (Ainsworth et al., 2023; Crisostomi et al., 2024), the performance decreases when the multiple models are merged.

- **Matching methods for multiple models.** We propose a permutation search method using a straight-through estimator for multiple models (STE-MM). In STE-MM, permutations are explored by repeatedly solving the linear assignment problem (LAP), which incurs a high computational cost. To address this, we also propose a method to accelerate the LAP solver.

- **Effectiveness verification of STE-MM through experiments and loss sharpness analysis.** We evaluate STE-MM on MLP, VGG-11, and ResNet-20 trained on MNIST, FMNIST, and CIFAR-10. Our results show that STE-MM enables the merging of multiple models while maintaining test accuracy and loss comparable to the originals. Additionally, the merged models exhibit reduced loss sharpness, which decreases further as more models are merged. These findings suggest that STE-MM facilitates the transfer of multiple models into an approximately convex loss basin.

## 2. Background

### 2.1. Notation

For any $k \in \mathbb{N}$, let $[k] = \{1, 2, \ldots, k\}$. Bold uppercase and lowercase variables represent matrices (e.g., $\boldsymbol{X}$), and vectors (e.g., $\boldsymbol{x}$), respectively. For any matrix $\boldsymbol{X}$, $\mathrm{vec}(\boldsymbol{X})$ denotes its vectorization, and $\|\boldsymbol{X}\|$ denotes its Frobenius ($L^2$) norm.

### 2.2. Permutation Invariance

For simplicity, we consider $L$-layer multilayer perceptrons (MLPs) $f(\boldsymbol{x}; \boldsymbol{\theta})$ while our analyses are applicable to any model architecture. Here, $\boldsymbol{x} \in \mathbb{R}^{d_{\mathrm{in}}}$ is an input to the NN, and $\boldsymbol{\theta} \in \mathbb{R}^{d_{\mathrm{param}}}$ represents the model parameters. Regarding the $\ell$-th layer output $\boldsymbol{z}_\ell$, we have $\boldsymbol{z}_0 = \boldsymbol{x}$, and, for all $\ell \in [L]$, $\boldsymbol{z}_\ell = \sigma(\boldsymbol{W}_\ell \boldsymbol{z}_{\ell-1} + \boldsymbol{b}_\ell)$, where $\sigma$ denotes the activation function, and $\boldsymbol{W}_\ell$ and $\boldsymbol{b}_\ell$ denote the $\ell$-th layer's weight and bias, respectively. Note that, in this MLP, the model parameters are given by $\boldsymbol{\theta} = \|_{\ell=1}^{L} (\mathrm{vec}(\boldsymbol{W}_\ell) \| \boldsymbol{b}_\ell)$, where $\|$ represents the concatenation of vectors. Neural networks (NNs) have permutation symmetries within their weight space. Let us consider an NN with model parameters $\boldsymbol{\theta}$. For the $\ell$-th layer, the following equation holds: $\boldsymbol{z}_\ell = \boldsymbol{P}^\top \boldsymbol{P} \boldsymbol{z}_\ell = \boldsymbol{P}^\top \sigma(\boldsymbol{P} \boldsymbol{W}_\ell \boldsymbol{z}_{\ell-1} + \boldsymbol{P} \boldsymbol{b}_\ell)$, where $\boldsymbol{P}$ represents a permutation matrix. Since permutation matrices are orthogonal, by permuting the input to the $(\ell+1)$-st layer using $\boldsymbol{P}^\top$, the model's parameters can be transformed without affecting the NN's input-output functionality. Specifically, the updated weights and bias are expressed as $\boldsymbol{W}_\ell' = \boldsymbol{P} \boldsymbol{W}_\ell$, $\boldsymbol{b}_\ell' = \boldsymbol{P} \boldsymbol{b}_\ell$, and $\boldsymbol{W}_{\ell+1}' = \boldsymbol{W}_{\ell+1} \boldsymbol{P}^\top$. This type of permutation can be applied independently to each layer. We define the tuple of permutation matrices applied across all layers as $\pi = (\boldsymbol{P}_\ell)_{\ell \in [L]}$. Furthermore, for a given model $\boldsymbol{\theta}$, applying a permutation $\pi$ to $\boldsymbol{\theta}$ is denoted as $\pi(\boldsymbol{\theta})$.

### 2.3. Permutation Selection For Two Models

Ainsworth et al. (2023) introduced two methods, weight matching (WM) and the straight-through estimator (STE), to identify permutations which satisfy Conjecture 1.1.

**Weight matching.** The WM aims to find a permutation $\pi$ that minimizes the $L_2$ distance between two models[1]:

$$\|\boldsymbol{\theta}_a - \pi(\boldsymbol{\theta}_b)\|^2 = \sum_{\ell \in [L]} \|\boldsymbol{W}_\ell^{(a)} - \boldsymbol{P}_\ell \boldsymbol{W}_\ell^{(b)} \boldsymbol{P}_{\ell-1}^\top\|^2, \quad (1)$$

where, without loss of generality, $\boldsymbol{P}_L = \boldsymbol{I}$ and $\boldsymbol{P}_0 = \boldsymbol{I}$, with $\boldsymbol{I}$ denoting the identity matrix. Ainsworth et al. (2023) propose a method to approximately solve this optimization by iteratively solving the linear assignment problem (LAP) because this optimization problem is NP-hard (Koopmans &

---

[1]Although the focus is on weights, biases can also be incorporated by concatenating them with the weights.

Beckmann, 1957; Sahni & Gonzalez, 1976; Ainsworth et al., 2023). We will describe their algorithm in Section 4.2.

**Straight-through estimator.** STE directly minimizes the loss barrier between the two models. Its objective function is expressed as $\arg\min_\pi \mathcal{L}\left(\frac{1}{2}\left(\boldsymbol{\theta}_a + \pi(\boldsymbol{\theta}_b)\right)\right)$. Since solving this directly is difficult, Ainsworth et al. (2023) proposed a strategy that splits the optimization into two stages: a forward pass to determine permutations and a backward pass to train the model using the obtained permutations. By iteratively repeating these steps, they approximated the solution effectively.

### 2.4. Linear Assignment Problem

The linear assignment problem (LAP) is used to determine the permutation matrix in permutation search methods such as WM and STE. Given a cost matrix $\boldsymbol{C}$ and a permutation matrix $\boldsymbol{P}$, the LAP can be formulated as follows:

$$
\begin{aligned}
\text{minimize} \quad & \sum_{i,j} \boldsymbol{C}_{i,j} \boldsymbol{P}_{i,j}, \\
\text{subject to} \quad & \sum_i \boldsymbol{P}_{i,j} = 1, \forall j \in [m], \\
& \sum_j \boldsymbol{P}_{i,j} = 1, \forall i \in [m], \\
& \boldsymbol{P}_{i,j} \geq 0, \forall i,j \in [m], \quad (2)
\end{aligned}
$$

where $m$ denotes the size of the matrices $\boldsymbol{C}$ and $\boldsymbol{P}$, and $\boldsymbol{C}_{i,j}$ and $\boldsymbol{P}_{i,j}$ represent the $(i,j)$-th components of matrices $\boldsymbol{C}$ and $\boldsymbol{P}$, respectively.

Classical algorithms such as the Hungarian algorithm and the Jonker-Volgenant algorithm (Kuhn, 1955; Jonker & Volgenant, 1987) reformulate the problem into the following dual problem to indirectly obtain the optimal solution:

$$
\begin{aligned}
\text{maximize} \quad & \sum_i \boldsymbol{v}_i + \sum_j \boldsymbol{u}_j, \\
\text{subject to} \quad & \boldsymbol{v}_i + \boldsymbol{u}_j \leq \boldsymbol{C}_{i,j}, \forall i,j \in [m], \quad (3)
\end{aligned}
$$

where $\boldsymbol{u} \in \mathbb{R}^m$ and $\boldsymbol{v} \in \mathbb{R}^m$ are dual variables.

According to the complementary slackness theorem of linear programming, the necessary and sufficient condition for the feasible solutions of the primal and dual problems to be optimal solutions is given by:

$$
\boldsymbol{P}_{i,j}(\boldsymbol{C}_{i,j} - \boldsymbol{v}_i - \boldsymbol{u}_j) = 0, \forall i,j \in [m]. \quad (4)
$$

From this complementary condition, if the optimal solutions $\boldsymbol{v}^*$ and $\boldsymbol{u}^*$ of the dual problem Equation (3) are obtained, it is possible to determine which components of the permutation matrix $\boldsymbol{P}$ must be zero. Using this information, the optimal solution of Equation (2) can be determined.

Equation (4) shows that adding or subtracting a constant to any row or column of the cost matrix $\boldsymbol{C}$ does not change the optimal solution (permutation matrix) of the primal problem. This property will be utilized in Section 4.2.

## 3. Difficulty in Transferring Multiple Models into Convex Loss Basin

Ainsworth et al. (2023); Entezari et al. (2022) have claimed that SGD solutions can be transferred into a single approximately convex loss basin by identifying a permutation that satisfies LMC between two SGD solutions. They experimentally demonstrated the existence of such a permutation when the model width is sufficiently large for MLP, VGG, and ResNet models trained on MNIST and CIFAR10.

However, even if this experimental observation holds generally, it does not necessarily imply that SGD solutions can be transferred into a single convex basin. For instance, consider three SGD solutions $\boldsymbol{\theta}_a$, $\boldsymbol{\theta}_b$, and $\boldsymbol{\theta}_c$. Assume that there exist permutations such that LMC holds for every model pair. In this case, there exist permutations $\pi_b$ and $\pi_c$ such that LMC holds between $\boldsymbol{\theta}_a$ and $\pi_b(\boldsymbol{\theta}_b)$, as well as between $\boldsymbol{\theta}_a$ and $\pi_c(\boldsymbol{\theta}_c)$. If these parameters can be transferred into the same convex basin, the loss within the triangle formed by $\boldsymbol{\theta}_a$, $\pi_b(\boldsymbol{\theta}_b)$, and $\pi_c(\boldsymbol{\theta}_c)$ must be uniformly low. However, there is no guarantee that $\pi_b(\boldsymbol{\theta}_b)$ and $\pi_c(\boldsymbol{\theta}_c)$ are linearly mode-connected. Note that, from the assumption, there should exist a permutation $\pi$ such that LMC holds between $\pi_b(\boldsymbol{\theta}_b)$ and $\pi(\pi_c(\boldsymbol{\theta}_c))$, but this does not necessarily mean LMC holds between $\pi_b(\boldsymbol{\theta}_b)$ and $\pi_c(\boldsymbol{\theta}_c)$. Therefore, it remains unclear whether these parameters can be gathered into a single convex basin. In this section, we experimentally demonstrate that as the number of models increases, the loss of the merged model increases, revealing that matching algorithms only for two models are insufficient.

### 3.1. Experimental Results

We experimentally demonstrate that matching algorithms between two models are insufficient for gathering multiple SGD solutions into a single convex basin. Specifically, we first prepare $n$ SGD solutions $\boldsymbol{\theta}_1, \boldsymbol{\theta}_2, \ldots, \boldsymbol{\theta}_n$. Next, for all $i \in [n-1]$, we identify a permutation $\pi_i$ such that LMC holds between $\boldsymbol{\theta}_n$ and $\pi_i(\boldsymbol{\theta}_i)$. Then, we investigate the test loss and accuracy of the merged model of $n-1$ models $\boldsymbol{\theta}_1, \ldots, \boldsymbol{\theta}_{n-1}$ after applying the permutations, $\frac{1}{n-1}\sum_{i \in [n-1]} \pi_i(\boldsymbol{\theta}_i)$. If the $n$ SGD solutions can be transferred into a single convex basin using the permutations $\pi_1, \pi_2, \ldots, \pi_{n-1}$, it is expected that the test loss and accuracy of these merged models would remain unchanged when the number of models used for model merging increases.

Figure 1 shows the experimental results. In this experiment, model merging was performed on ResNet-20 and VGG-11

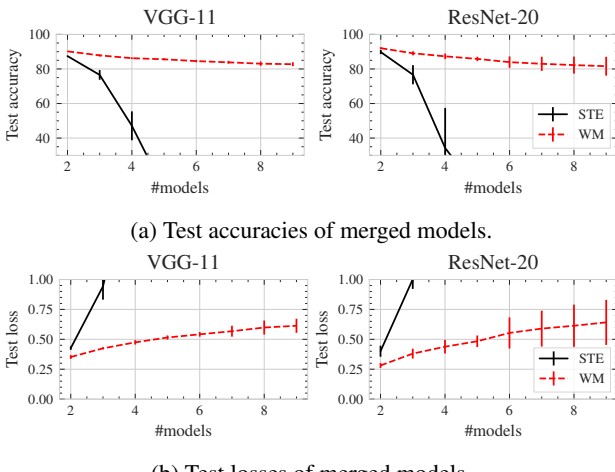

(a) Test accuracies of merged models.

(b) Test losses of merged models.

Figure 1: Test accuracy and loss of the merged model $\frac{1}{n-1}\sum_{i\in[n-1]}\pi_i(\boldsymbol{\theta}_i)$ when the number of models used for model merging is increased. The figures show the mean and standard deviation from three trials of model merging.

models trained on the CIFAR10 dataset. Details on the dataset used, the training procedures for the models, and the experimental conditions for model merging, are described in Appendix B. Similar to prior studies (Ainsworth et al., 2023; Yamada et al., 2024), the model width is increased to establish LMC between two models (specifically, by 16 times for ResNet-20 and 4 times for VGG-11).

Figure 1 shows that both STE and WM result in a monotonic degradation of the test accuracy and loss of the merged model as the number of models increases. Notably, STE exhibits a sharp decline in performance. These results indicate that the conventional methods using WM and STE cannot transfer all models into the same convex loss basin.

## 4. Permutation Search for Multiple Models

In this section, we propose STE for multiple models (STE-MM) that handle multiple models simultaneously to transfer multiple SGD solutions into a single convex loss basin. First, in Section 4.1, we explain the algorithm of STE-MM. STE-MM is derived by extending STE to multiple models and, therefore, uses SGD to find permutations. However, this training process becomes computationally expensive since solving the linear assignment problem (LAP) is required at each iteration of SGD. To address this issue, in Section 4.2, we propose a method to accelerate the LAP solving process.

### 4.1. Straight-through Estimator for Multiple Models

Merging multiple independently trained models is challenging because it requires solving a combinatorial optimization problem over permutations. Inspired by the success

---

**Algorithm 1** STE-MM

**Require:** Models $\boldsymbol{\theta}_1, \boldsymbol{\theta}_2, \ldots, \boldsymbol{\theta}_n$, the number of epochs $n_e$
**Ensure:** Permutations $\pi_1, \ldots, \pi_{n-1}$
1: $\hat{\boldsymbol{\theta}}_i \leftarrow \boldsymbol{\theta}_n$ for all $i \in [n-1]$      ▷ Initialize dummy parameters $\hat{\boldsymbol{\theta}}_i$.
2: **for** $e = 1 \ldots N_e$ **do**
3:      $\pi_i' \leftarrow \texttt{WM}(\hat{\boldsymbol{\theta}}_i, \boldsymbol{\theta}_i)$ for all $i \in [n-1]$
4:      $(\lambda_i)_{i=1}^{n} \leftarrow \texttt{UniformDist}((0,1))$
5:      $\lambda_i \leftarrow \lambda_i / \sum_j \lambda_j$ for all $i \in [n]$
6:      $\text{loss} \leftarrow \mathcal{L}(\sum_{i=1}^{n-1} \lambda_i \pi_i'(\boldsymbol{\theta}_i) + \lambda_n \boldsymbol{\theta}_n)$
7:      **for** $i = 1 \ldots n-1$ **do**
8:          $\text{grad}_i \leftarrow \texttt{GetGradient}(\text{loss}, \pi_i'(\boldsymbol{\theta}_i))$
9:          $\hat{\boldsymbol{\theta}}_i \leftarrow \texttt{Update}(\hat{\boldsymbol{\theta}}_i, \text{grad}_i)$
10:      **end for**
11: **end for**
12: $\pi_i \leftarrow \texttt{WM}(\hat{\boldsymbol{\theta}}_i, \boldsymbol{\theta}_i)$ for all $i \in [n-1]$

---

of straight-through estimators (STEs) in other discrete optimization problems (Bengio et al., 2013; Ainsworth et al., 2023), we propose a straight-through estimator for multiple models (STE-MM), a method to learn permutations that can transfer multiple models into the same convex loss basin in a differentiable framework.

Let $\boldsymbol{\theta}_1, \boldsymbol{\theta}_2, \ldots, \boldsymbol{\theta}_n$ be model parameters obtained by SGD. Our goal is to find permutations $\pi_1, \ldots, \pi_{n-1}$ such that their convex combination minimizes the expected loss:

$$\underset{\pi_1, \ldots, \pi_{n-1}}{\arg\min} \ \mathbb{E}_{\lambda_1, \ldots, \lambda_n} \mathcal{L}\left(\sum_{i=1}^{n-1} \lambda_i \pi_i(\boldsymbol{\theta}_i) + \lambda_n \boldsymbol{\theta}_n\right), \quad (5)$$

where $\lambda_1, \ldots, \lambda_n$ are sampled from a uniform distribution over the simplex (i.e., positive and summing to 1). We optimize only $n-1$ permutations, since the last one can be fixed without loss of generality thanks to Theorem D.1.

Direct optimization of Equation (5) is infeasible because permutations are discrete and non-differentiable. To address this, we adopt a STE approach: instead of optimizing permutations directly, we introduce dummy parameters $\hat{\boldsymbol{\theta}}_1, \ldots, \hat{\boldsymbol{\theta}}_{n-1}$ as continuous proxies. During the forward pass, we compute the best permutation $\pi_i'$ aligning $\pi_i'(\boldsymbol{\theta}_i)$ to $\hat{\boldsymbol{\theta}}_i$ using the WM algorithm from (Ainsworth et al., 2023). In the backward pass, we compute gradients with respect to $\pi_i'(\boldsymbol{\theta}_i)$ and update $\hat{\boldsymbol{\theta}}_i$ via standard gradient descent.

This procedure is formalized in Algorithm 1. At initialization (line 1), we set all $\hat{\boldsymbol{\theta}}_i = \boldsymbol{\theta}_n$ as suggested by prior work, since it improves the quality of merged models. Then, for each training epoch:

- (Line 3) For each $i$, find the permutation $\pi_i'$ minimizing $\|\hat{\boldsymbol{\theta}}_i - \pi_i'(\boldsymbol{\theta}_i)\|^2$ using the WM algorithm.

- (Lines 4–5) Sample mixture weights $\lambda_1, \ldots, \lambda_n$ from a uniform simplex.

- (Line 6) Compute the loss of the weighted sum of permuted models.

- (Lines 7–10) Compute gradients w.r.t. $\pi_i'(\boldsymbol{\theta}_i)$ and update $\hat{\boldsymbol{\theta}}_i$ accordingly.

After training, the final permutations $\pi_i$ are obtained by aligning the updated $\hat{\boldsymbol{\theta}}_i$ with $\boldsymbol{\theta}_i$ again via WM (line 12).

The key insight behind this approach is that if $\pi_i'(\boldsymbol{\theta}_i)$ becomes sufficiently close to $\hat{\boldsymbol{\theta}}_i$ during training (i.e., $\pi_i'(\boldsymbol{\theta}_i) \approx \hat{\boldsymbol{\theta}}_i$), then $\hat{\boldsymbol{\theta}}_i$ can be regarded as a good approximation to $\pi_i'(\boldsymbol{\theta}_i)$. This idea is supported by empirical results (Section 5), showing that our method consistently transfers models into a shared low-loss region, suggesting an approximate convexity in the parameter space under learned permutations.

## 4.2. Accelerating WM

This subsection proposes a method to accelerate WM to speed up STE-MM. In particular, since STE-MM requires performing WM in every iteration of each epoch, this acceleration leads to a significant improvement in efficiency. To explain the proposed method, we first describe the original WM algorithm. Next, we present the fundamental idea for solving the LAP (linear assignment problem) more quickly. Then, we describe the WM algorithm incorporating the proposed method. Finally, we show the experimental results of our proposed method to verify its effectiveness.

**Algorithm of weight matching.** For two given models $\boldsymbol{\theta}_a$ and $\boldsymbol{\theta}_b$, the goal of WM is to find a permutation $\pi$ that minimizes $\|\boldsymbol{\theta}_a - \pi(\boldsymbol{\theta}_b)\|^2$. Ainsworth et al. (2023) proposed Algorithm 2 as a method to approximately solve Equation (1). First, in line 1, the permutation matrices are initialized as identity matrices $(\boldsymbol{I}_{n_\ell})_{\ell=0}^L$, where $n_\ell$ represents the size of the permutation matrix for $\ell$-th layer. Next, the permutation matrices are updated in lines 3–6 until the $L^2$ distance $\|\boldsymbol{\theta}_a - \pi(\boldsymbol{\theta}_b)\|$ between the two models converges. In line 3, a list of permutation matrices to be updated is prepared. Then, in line 5, a layer is randomly selected from this list, and it is removed from the list. In line 6, the cost matrix $\boldsymbol{C}$ corresponding to the selected layer $\ell$ is computed, and in line 7, the permutation matrix $\boldsymbol{P}_\ell$ that minimizes this cost is obtained by solving LAP. The obtained permutation matrix $\boldsymbol{P}_\ell$ minimizes the following term: $\|\boldsymbol{W}_{\ell+1}^{(a)} - \boldsymbol{P}_{\ell+1}\boldsymbol{W}_{\ell+1}^{(b)}\boldsymbol{P}_\ell^\top\|^2 + \|\boldsymbol{W}_\ell^{(a)} - \boldsymbol{P}_\ell\boldsymbol{W}_\ell^{(b)}\boldsymbol{P}_{\ell-1}^\top\|^2$. The most time-consuming step in WM is solving the LAP. For instance, the computational cost of solving LAP with the Hungarian algorithm is $O(n_\ell^3)$. Therefore, we propose a method to solve LAP more efficiently.

---

**Algorithm 2** WM

**Require:** Two SGD solutions $\boldsymbol{\theta}_a = \big\|_{\ell=1}^L (\mathrm{vec}(\boldsymbol{W}_\ell^{(a)}))$ and
$\boldsymbol{\theta}_b = \big\|_{\ell=1}^L (\mathrm{vec}(\boldsymbol{W}_\ell^{(b)}))$
**Ensure:** Permutation $\pi$
1: $\pi \leftarrow (\boldsymbol{I}_{n_\ell})_{\ell=0}^L$
2: **while** $\|\boldsymbol{\theta}_a - \pi(\boldsymbol{\theta}_b)\|$ has not converged **do**
3: $\quad$ LayerList $\leftarrow \{1, 2, \ldots, L-1\}$
4: $\quad$ **while** LayerList is not empty **do**
5: $\quad\quad \ell \leftarrow \texttt{PopRandomly}(\text{LayerList})$
6: $\quad\quad \boldsymbol{C} \leftarrow -\boldsymbol{W}_\ell^{(a)}\boldsymbol{P}_{\ell-1}\boldsymbol{W}_\ell^{(b)\top} - \boldsymbol{W}_{\ell+1}^{(a)\top}\boldsymbol{P}_{\ell+1}\boldsymbol{W}_{\ell+1}^{(b)}$
7: $\quad\quad \boldsymbol{P}_\ell \leftarrow \texttt{SolveLAP}(\boldsymbol{C})$
8: $\quad$ **end while**
9: **end while**

---

**Basic idea.** The idea to speed up the LAP solver is to use the optimal solution of the dual problem obtained in the previous iteration to adjust the cost matrix. Below, we provide an intuitive explanation of this idea. In the **while** loop on lines 2–9 of Algorithm 2, let $\boldsymbol{C}'$, $\boldsymbol{P}'$, $\boldsymbol{v}'$, and $\boldsymbol{u}'$ be the cost matrix, optimal permutation matrix, and dual variables for the $\ell$-th layer in a given iteration, respectively. Similarly, let $\boldsymbol{C}$ and $\boldsymbol{P}$ be the cost matrix and optimal permutation matrix in the next iteration. For simplicity, we omit the subscript $\ell$ in the variables. If the iterations have sufficiently progressed, the permutation in Algorithm 2 is expected to converge. Thus, we can assume that $\boldsymbol{C}' \approx \boldsymbol{C}$ and $\boldsymbol{P}' \approx \boldsymbol{P}$. Complementary slackness condition yields that $0 = \boldsymbol{P}_{i,j}'(\boldsymbol{C}_{i,j}' - \boldsymbol{v}_i' - \boldsymbol{u}_j') \approx \boldsymbol{P}_{i,j}(\boldsymbol{C}_{i,j} - \boldsymbol{v}_i' - \boldsymbol{u}_j')$. Based on this, we define a new cost matrix $\boldsymbol{C}_{i,j}^{(\mathrm{new})} = \boldsymbol{C}_{i,j} - \boldsymbol{v}_i' - \boldsymbol{u}_j'$ and solve the LAP for $\boldsymbol{C}^{(\mathrm{new})}$ instead of $\boldsymbol{C}$. Note that the optimal solution (permutation) does not change when transitioning from $\boldsymbol{C}$ to $\boldsymbol{C}^{(\mathrm{new})}$. Let $\boldsymbol{v}$ and $\boldsymbol{u}$ be the optimal solutions to the dual problem for $\boldsymbol{C}^{(\mathrm{new})}$. From Equation (4), $\boldsymbol{P}_{i,j}(\boldsymbol{C}_{i,j}^{(\mathrm{new})} - \boldsymbol{v}_i - \boldsymbol{u}_j) = \boldsymbol{P}_{i,j}(\boldsymbol{C}_{i,j} - \boldsymbol{v}_i' - \boldsymbol{u}_j' - \boldsymbol{v}_i - \boldsymbol{u}_j) = 0$ must hold. However, if $\boldsymbol{P}_{i,j}(\boldsymbol{C}_{i,j} - \boldsymbol{v}_i' - \boldsymbol{u}_j') \approx 0$ holds, then $\boldsymbol{v}_i$ and $\boldsymbol{u}_j$ are expected to be close to zero, even without explicitly solving for them. Therefore, solving the dual problem for $\boldsymbol{C}^{(\mathrm{new})}$ is likely to be easier than for $\boldsymbol{C}$.

**Proposed method.** The accelerated WM algorithm is shown in Algorithm 3. Since most of Algorithm 3 is the same as Algorithm 2, we will only explain the differences between them. The first difference is that Algorithm 3 takes the dual variables $(\boldsymbol{v}_\ell)_\ell$ and $(\boldsymbol{u}_\ell)_\ell$ as inputs and also returns the dual variables obtained at the end of the algorithm. This is used to accelerate STE. Specifically, in Algorithm 1, WM function is replaced with the proposed accelerated WM algorithm, and the dual variables of the $i$-th model obtained from the previous epoch (or the previous iteration of SGD for finding permutation) are passed to the replaced WM func-

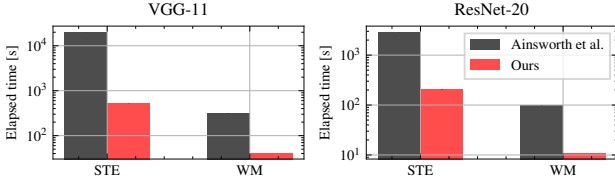

Figure 2: Processing time of WM and STE.

---

**Algorithm 3** Accelerated WM

---

**Require:** Two SGD solutions $\boldsymbol{\theta}_a$ and $\boldsymbol{\theta}_b$, dual variables $(\boldsymbol{v}_\ell)_\ell$ and $(\boldsymbol{u}_\ell)_\ell$
**Ensure:** Permutation $\pi$, dual variables $(\boldsymbol{v}_\ell)_\ell$ and $(\boldsymbol{u}_\ell)_\ell$
 1: $\pi \leftarrow (\boldsymbol{I}_{n_\ell})_{\ell=0}^{L}$
 2: **while** $\|\boldsymbol{\theta}_a - \pi(\boldsymbol{\theta}_b)\|$ has not converged **do**
 3:     LayerList $\leftarrow \{1, 2, \ldots, L-1\}$
 4:     **while** LayerList is not empty **do**
 5:         $\ell \leftarrow \texttt{PopRandomly}(\text{LayerList})$
 6:         $\boldsymbol{C} \leftarrow -\boldsymbol{W}_\ell^{(a)}\boldsymbol{P}_{\ell-1}\boldsymbol{W}_\ell^{(b)^\top} - \boldsymbol{W}_{\ell+1}^{(a)^\top}\boldsymbol{P}_{\ell+1}\boldsymbol{W}_{\ell+1}^{(b)}$
 7:         $(\boldsymbol{C}_{i,j}') \leftarrow (\boldsymbol{C}_{i,j} - \boldsymbol{u}_{\ell,i} - \boldsymbol{v}_{\ell,j})$
 8:         $\boldsymbol{P}_\ell, \boldsymbol{u}_\ell', \boldsymbol{v}_\ell' \leftarrow \texttt{SolveLAPWithDuals}(\boldsymbol{C}')$
 9:         $\boldsymbol{v}_\ell \leftarrow \boldsymbol{v}_\ell + \boldsymbol{v}_\ell'; \boldsymbol{u}_\ell \leftarrow \boldsymbol{u}_\ell + \boldsymbol{u}_\ell'$
10:     **end while**
11: **end while**

---

tion, enabling faster execution. In fact, our implementation of STE-MM adopts this approach. Note that, when the first iteration of STE-MM is executed, $u_\ell$ and $v_\ell$ input to the WM are set to $\boldsymbol{0}$. The second difference is in line 7, where the dual variables are subtracted from the cost matrix to get $\boldsymbol{C}^{(\text{new})}$. This step has already been explained above. The third difference is in line 8, where not only the permutation matrix but also the dual variables $\boldsymbol{u}_\ell'$ and $\boldsymbol{v}_\ell'$ obtained by solving the LAP are returned. Then, in line 9, the dual variables $\boldsymbol{v}_\ell$ and $\boldsymbol{u}_\ell$ are updated by adding these values. By using this adjustment, the dual variables $\boldsymbol{v}_\ell$ and $\boldsymbol{u}_\ell$ obtained in line 9 match the optimal solutions of the dual problem for the original cost matrix $\boldsymbol{C}$.

**Experimental comparison.** Figure 2 shows the processing time of WM and STE to find permutations for two models, with and without the proposed acceleration of the LAP solver described in Section 4.2. The figure presents the mean and standard deviation of processing time when a permutation search is conducted three times. For STE, the number of epochs was set to one. From the figure, we can see that both WM and STE with our acceleration method are at least approximately ten times faster than those without it, regardless of the model architecture. In particular, when finding permutations for ResNet-20 models, our method reduces the processing time to $1/40$ of the original.

## 5. Merging Multiple Models

This section conducts some experiments using multiple SGD solutions to confirm the effectiveness of STE-MM. Since verifying directly that all models belong to a convex basin is difficult, we investigate the test losses of merged models within a scope that can be easily verified. In the next section, a deeper analysis is conducted, focusing on the sharpness of the loss function around the merged models.

Section 5.1 first examines the losses of merged models obtained by applying permutations. Then, in Section 5.2, we examine the losses between all pairs of models. These losses are expected to be comparable to the original models. Finally, Section 5.3 experimentally demonstrates that larger model widths lead to lower losses in the merged models.

### 5.1. Merging All Models

Figure 3(a) shows the test accuracy of the merged model. Specifically, given $n$ SGD solutions $\boldsymbol{\theta}_1, \boldsymbol{\theta}_2, \ldots, \boldsymbol{\theta}_n$ and the coresponding permutations $\pi_1, \pi_2, \ldots, \pi_n$ (where, in STE-MM, $\pi_n = (\boldsymbol{I}_{n_\ell})_{\ell \in [L]}$ holds), it presents the test accuracy of the merged model $\frac{1}{n}\sum_i \pi_i(\boldsymbol{\theta}_i)$. For reference, the figure also includes results using two other methods for model merging: MergeMany, proposed in Appendix A.10 of Ainsworth et al. (2023), and cycle consistent multi-model merging (CCMM), proposed by Crisostomi et al. (2024). Both methods aim to find permutations $\pi_1, \pi_2, \ldots, \pi_n$ that minimize the sum of $L^2$ distances between all model pairs, $\sum_{i,j} \|\pi_i(\boldsymbol{\theta}_i) - \pi_j(\boldsymbol{\theta}_j)\|^2$. Additionally, Figure 3(b) shows how much the test loss of the merged model increases compared to the original models' losses.

First, Figure 3(a) shows that the test accuracy of the merged model increases with the number of models with STE-MM, whereas it decreases with other methods. This difference is likely due to the fact that the objective function of STE-MM directly minimizes the loss values, whereas the objective functions of MergeMany and CCMM focus on reducing the distances between models. Next, Figure 3(b) shows that, except for the MLP models trained on MNIST, increasing the number of models leads to an increase in the test loss. Notably, the increase is more pronounced in MergeMany and CCMM. While the loss of the merged model also tends to increase with the number of models in STE-MM, the increase in test loss value becomes small as the number of models grows. In fact, as shown in Figure 3(a), the test accuracy improves as the number of models increases, suggesting that the increase in test loss may converge. Furthermore, the difference between the original models' losses and the merged model's loss is close to zero in STE-MM, indicating that the merged model's loss is sufficiently small.

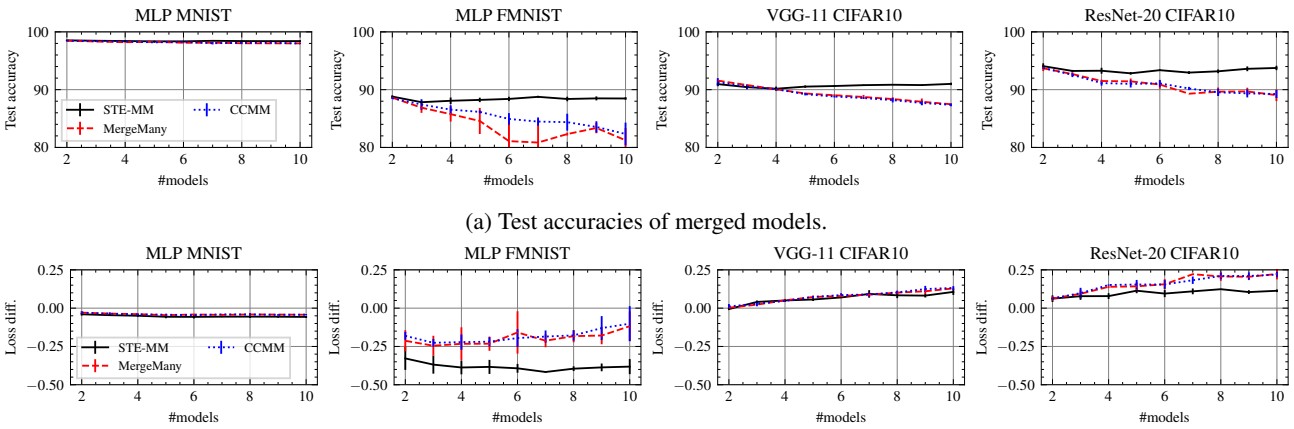

(a) Test accuracies of merged models.

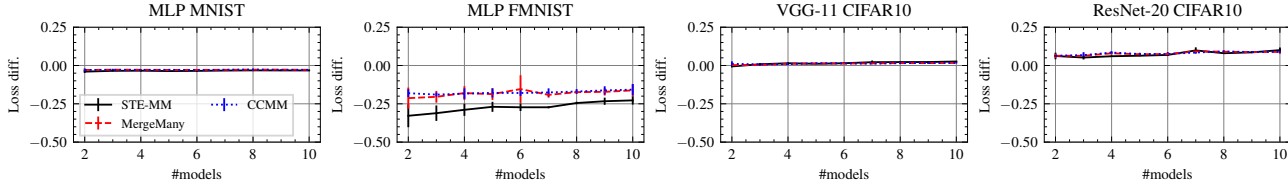

(b) Difference between the test loss of merged model and the average test loss of original models: $\mathcal{L}(\frac{1}{n}\sum_i \pi_i(\boldsymbol{\theta}_i)) - \frac{1}{n}\sum_i \mathcal{L}(\boldsymbol{\theta}_i)$.

Figure 3: Experimental results of merged models $\frac{1}{n}\sum_{i\in[n]}\pi_i(\boldsymbol{\theta}_i)$ when the number of models is increased. The figures show the mean and standard deviation from three trials of model merging.

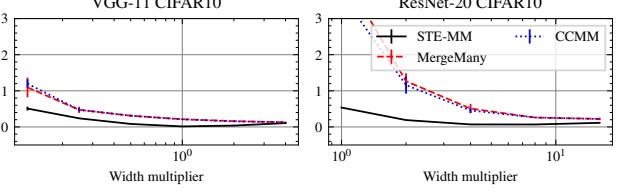

Figure 4: Difference in average test losses between the merged models (for all model pairs) and the original models: $\frac{2}{n(n-1)}\sum_{i<j}\mathcal{L}(\frac{1}{2}(\pi_i(\boldsymbol{\theta}_i) + \pi_j(\boldsymbol{\theta}_j))) - \frac{1}{n}\sum_i \mathcal{L}(\boldsymbol{\theta}_i)$.

## 5.2. Merging Each Model Pair

Next, we investigate the loss when merging each model pair. Figure 4 illustrates the difference between the average loss of the merged models for all model pairs and the loss of the original models (i.e., $\frac{2}{n(n-1)}\sum_{i<j}\mathcal{L}(\frac{1}{2}(\pi_i(\boldsymbol{\theta}_i) + \pi_j(\boldsymbol{\theta}_j))) - \frac{1}{n}\sum_i \mathcal{L}(\boldsymbol{\theta}_i)$). For MLP models trained on FM-NIST, it was observed that the average loss of the model pairs increases when using STE-MM. However, the difference in loss remains below 0, and the amount of the increase in loss value diminishes as the number of models grows. This suggests that increasing the number of models may not pose an issue when transferring multiple models to the same low-loss basin. For MLP, VGG-11, and ResNet-20 models trained on MNIST or CIFAR-10, the impact of increasing the number of models on the loss is very small, with the difference in loss remaining very close to 0. Consequently, this indicates that the loss barrier between any model pair is not particularly significant.

## 5.3. Model Width

Finally, we examine the impact of model width on the test loss of the merged model. Figure 5 shows the effect of model width on the difference between the loss of the

Figure 5: Difference between the test loss of the merged model and the average test loss of the original models when varying model width.

merged model of all the original models and the average loss of the original models when ten SGD solutions are merged. The figure shows that, for all methods, the wider the model, the closer the loss difference approaches zero. This result is the same as the trend observed in previous studies on merging two models, indicating that increasing the model width also facilitates LMC in multiple models.

## 6. Sharpness of Merged Model

This section investigates the sharpness of the loss for the merged model. If the permuted models belong to the same low-loss convex basin, the sharpness of the merged model's

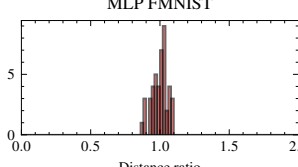 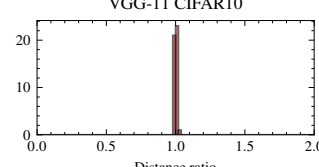 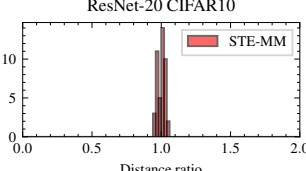

Figure 6: Histogram of $L^2$ distances of all model pairs using STE.

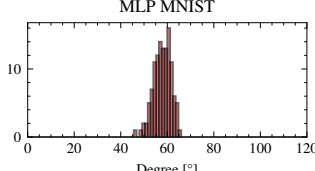 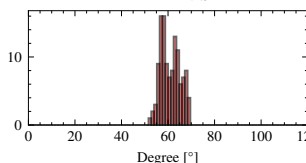 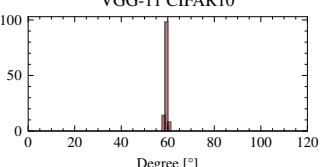 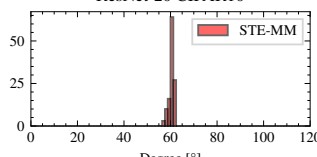

Figure 7: Histogram of degrees of all model pairs using STE.

loss can be expected to be smaller than that of the original models. First, in Section 6.1, we examine the geometric relation of models in the parameter space after applying the permutations and reveal that the models form an approximately regular simplex. Next, in Section 6.2, we measure sharpness using the Hessian matrix of the models and demonstrate that the sharpness of the merged model is smaller than that of the original models when using STE-MM.

## 6.1. Geometric Relation of Models

Figure 6 shows the $L^2$ distances between each pair of $n$ permuted models $\pi_1(\boldsymbol{\theta}_1), \pi_2(\boldsymbol{\theta}_2), \dots, \pi_n(\boldsymbol{\theta}_n)$ (i.e., $\|\pi_i(\boldsymbol{\theta}_i) - \pi_j(\boldsymbol{\theta}_j)\|$). This figure illustrates the results for $n = 10$ models when STE-MM is used to find permutations. The experimental results for other permutation methods are shown in Figure 9. Since there are $\frac{n(n-1)}{2} = 45$ model pairs, the $L^2$ distances are presented as a histogram.

The horizontal axis in Figure 9 represents the ratio of the $L^2$ distance for each model pair to the average $L^2$ distance across all pairs (i.e., $\|\pi_i(\boldsymbol{\theta}_i) - \pi_j(\boldsymbol{\theta}_j)\| / \frac{2}{n(n-1)} \sum_{i<j} \|\pi_i(\boldsymbol{\theta}_i) - \pi_j(\boldsymbol{\theta}_j)\|$). Figure 9 shows that the distances for all model pairs are close to the average distance. This tendency is particularly strong for VGG-11 and ResNet-20, where the distance ratios are very close to 1. This result suggests that the permuted models are located at approximately equal distances from each other, forming an $n$-dimensional regular simplex. Figures 7 and 10 also shows the angles between all possible triplets of models, which supports the conclusion that the models form a regular simplex.

## 6.2. Sharpness

Since the models $\pi_1(\boldsymbol{\theta}_1), \pi_2(\boldsymbol{\theta}_2), \dots, \pi_n(\boldsymbol{\theta}_n)$, after applying permutations, form an approximately regular simplex in

$n$ dimensions, the condition for these models to form a convex low-loss basin is that the losses at all points within the regular simplex are small. However, as the dimensionality increases, the space of points where loss needs to be verified grows exponentially, making it impractical to examine the loss at all points. Therefore, in this subsection, we investigate the sharpness of the loss at the center point, which corresponds to the merged model. If multiple models form a convex low-loss basin, it is expected that the sharpness of the loss at the center will be smaller than the sharpness at the vertices (i.e., the trained models).

**Definition of sharpness.** Although there are various definitions of sharpness, we focus on the trace of the Hessian matrix $\boldsymbol{H}$ of the loss function, $\operatorname{tr} \boldsymbol{H}$, which measures the sharpness of the loss in a random direction (Wen et al., 2023). $\operatorname{tr} \boldsymbol{H}$ has good theoretical properties. For example, under certain assumptions, sharpness-aware minimization (Foret et al., 2021) and label noise SGD induce a small trace of the Hessian (Bartlett et al., 2023; Damian et al., 2021; Wen et al., 2023). However, using $\operatorname{tr} \boldsymbol{H}$ directly presents an issue for rectifier neural networks (i.e., those using ReLU as an activation function), as its value is not unique (Li et al., 2018; Kwon et al., 2021). Specifically, $\operatorname{tr} \boldsymbol{H}$ is not invariant to weight rescaling, even though such rescaling does not change the input-output functionality. To address this issue, instead of $\operatorname{tr} \boldsymbol{H}$, we use $\operatorname{tr}(\boldsymbol{H} \odot \boldsymbol{\theta}\boldsymbol{\theta}^\top)$, which is known to asymptotically match the average-case sharpness (Andriushchenko et al., 2023). Experimentally, to compute $\operatorname{tr}(\boldsymbol{H} \odot \boldsymbol{\theta}\boldsymbol{\theta}^\top)$ for a model $\boldsymbol{\theta}$, we use the following equation:

$$\operatorname{tr}(\boldsymbol{H} \odot \boldsymbol{\theta}\boldsymbol{\theta}^\top) = \mathbb{E}_{\boldsymbol{z}}(\boldsymbol{z}\operatorname{diag}(\boldsymbol{\theta}))^\top \boldsymbol{H}(\boldsymbol{z}\operatorname{diag}(\boldsymbol{\theta})), \quad (6)$$

where $\boldsymbol{z}$ is a Rademacher random vector, and $\operatorname{diag}(\boldsymbol{\theta})$ denotes a diagonal matrix with the elements of $\boldsymbol{\theta}$ as its diagonal entries. The proof of this equation is provided in Appendix D.2. The product of any vector

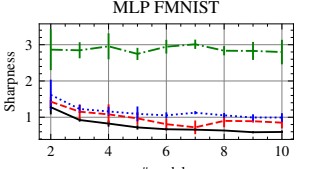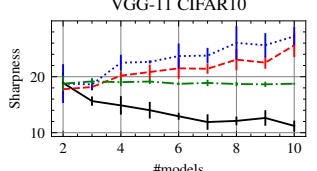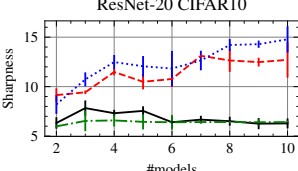

Figure 8: Sharpness of merged models.

$\mu$ and the Hessian matrix $H$ (i.e., $\mu H$) can be computed using `torch.autograd.functional.vhp` in PyTorch[2]. This enables the evaluation of $\text{tr}(H \odot \theta\theta^\top)$.

**Experimental results.** Figure 8 shows the sharpness of the merged models. In the figure, "Baseline" represents the average sharpness of the models used for merging. For MLP models trained on MNIST and FMNIST, Figure 8 shows that the sharpness decreases as the number of models increases, regardless of the methods. Notably, the sharpness of the merged models is smaller than that of the original models. On the other hand, for VGG-11 and ResNet-20 models, as the number of models increases, the sharpness monotonically increases for MergeMany and CCMM, while it decreases for STE-MM. This indicates that when models are transferred using the permutations found by STE-MM, the loss in the center of those models is, on average, flat. Thus, this indicates that STE-MM would successfully gather the models into an approximately convex low-loss basin.

## 7. Conclusion

In this paper, we analyzed whether multiple models can be gathered into a single approximately convex basin by using the permutation symmetries of neural networks. First, we experimentally showed that the performance of merged models decreases when the number of models is increased in the conventional permutation search method for two models. Next, we proposed a permutation search method using a straight-through estimator for multiple models (STE-MM). We then demonstrated the effectiveness of STE-MM using MLP, VGG-11, and ResNet-20 models trained on MNIST, FMNIST, and CIFAR10. Finally, we investigated the sharpness of the loss function and found that the sharpness of the merged model monotonically decreases when using STE-MM, which indicates that the multiple models are gathered into a single approximately convex basin.

These findings provide a promising foundation, but several important questions remain open for future work. This study focused on image classification tasks using relatively small

models and simple datasets. An important direction for future work is to investigate the applicability and effectiveness of the proposed method in more practical settings involving larger models and more complex data. Moreover, although STE-MM successfully discovers suitable permutations for merging multiple models, the reason for the existence of such permutations remains unclear. Clarifying this from the perspective of the learning dynamics of SGD would contribute to a deeper theoretical understanding. In addition, it is widely observed that satisfying LMC using permutation symmetries of NNs often requires increasing the model width. Exploring whether LMC can be achieved without expanding the width, potentially by developing more advanced permutation strategies or leveraging inherent architectural properties, remains an important open question.

## Impact Statement

This paper presents work whose goal is to advance the field of Machine Learning. There are many potential societal consequences of our work, none which we feel must be specifically highlighted here.

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

Information Processing Systems*, 2023. URL https:
//openreview.net/forum?id=vORUHrVEnH.

# A. Additional Related Work

**(Linear) Mode Connectivity.** Several studies (Garipov et al., 2018; Draxler et al., 2018; Freeman & Bruna, 2017) have demonstrated that different neural networks (NNs) can be connected through simple nonlinear paths with negligible loss increase. Nagarajan & Kolter (2019) were the first to discover that, in the case of models trained on MNIST with the same random initializations, solutions obtained via stochastic gradient descent (SGD) can also be connected by linear paths while maintaining an almost constant loss value. Subsequently, Frankle et al. (2020) empirically showed that linear mode connectivity (LMC) does not always hold between two SGD solutions, even when they share the same initialization, as it depends on the dataset and model architecture. However, they also demonstrated that if a model is initially trained for a certain duration and then used as the starting point for training two independent models, these models are linearly mode-connected. Furthermore, Frankle et al. (2020) explored the relationship between LMC and the lottery-ticket hypothesis (Frankle & Carbin, 2019). Entezari et al. (2022) conjectured that LMC holds, with high probability, when considering permutation symmetries in hidden layers. Later, Ainsworth et al. (2023) proposed a weight-matching (WM) method by formulating neuron alignment as a bipartite graph matching problem and solving it approximately. Following this, Peña et al. (2023) introduced the use of Sinkhorn's algorithm to directly solve the WM problem. Although several works (Venturi et al., 2019; Nguyen et al., 2019; Nguyen, 2019; Kuditipudi et al., 2019) have investigated nonlinear mode connectivity, theoretical analyses of LMC remain limited. Ferbach et al. (2024) provided an upper bound on the minimal width of the hidden layer to satisfy LMC if the independence of all neurons' weight vectors inside a given layer holds. Zhou et al. (2023) introduced the concept of layerwise linear feature connectivity (LLFC) and established that LLFC implies LMC. Ito et al. (2025) demonstrated that the top singular vectors of the parameters obtained by SGD play an important role in satisfying LMC, especially when permutations are found using WM. There are few studies on merging multiple models using permutations. Ainsworth et al. (2023) proposed MergeMany, which extends WM to multiple models. Crisostomi et al. (2024) proposed a method for searching permutations by minimizing the sum of distances between all model pairs as the objective function using the Frank-Wolfe algorithm. However, as shown in Section 5, these methods have the problem that the performance of the merged model worsens as the number of models increases.

**Model Merging.** Model merging and federated learning are relevant topics in the study of LMC. McMahan et al. (2017) and Konečný et al. (2016) are the first to introduce federated learning, a technique where models are trained on partitioned datasets. Wang et al. (2020) proposed a federated learning approach that involves permuting individual model components before averaging their weights. Similarly, Singh & Jaggi (2020) developed a model merging technique that aligns model weights using optimal transport, a method conceptually similar to that of Ainsworth et al. (2023). While Singh & Jaggi (2020)'s approach is designed for model fusion and performs worse than Ainsworth et al.'s method, it can still be categorized as an LMC-based technique due to its use of hard alignments within the same architecture. Wortsman et al. (2022) introduced a weight-averaging strategy that enhances test accuracy without increasing inference costs, distinguishing it from traditional ensemble methods.

# B. Experimental Settings

This section describes the experimental setup for training neural networks to obtain SGD solutions. In addition, we describe the hyperparameters for permutation search methods. Three datasets were used in this study: MNIST (Lecun et al., 1998), Fashion-MNIST (FMNIST) (Xiao et al., 2017), and CIFAR10 (Krizhevsky et al., 2009).

All experiments were conducted on a Linux workstation with two AMD EPYC 7543 32-Core processors, eight NVIDIA A30 GPUs, and 512 GB of memory. The PyTorch 2.5.1[3], PyTorch Lightning 2.4.0[4], and torchvision 0.20.1[5] libraries were used for model training and evaluation.

## B.1. Model Training

**MLP on MNIST and FMNIST.** Following the approach outlined in (Ainsworth et al., 2023), we implemented a multi-layer perceptron (MLP) consisting of three hidden layers, each with 512 units. The ReLU activation function was employed for the hidden layers. For training on the MNIST and FMNIST datasets, the Adam optimizer was utilized with a learning rate of $1 \times 10^{-3}$. The batch size was fixed at 512, and training was conducted for a maximum of 100 epochs. We did not

---

[3]https://pytorch.org/
[4]https://lightning.ai/docs/pytorch/stable/
[5]https://pytorch.org/vision/stable/index.html

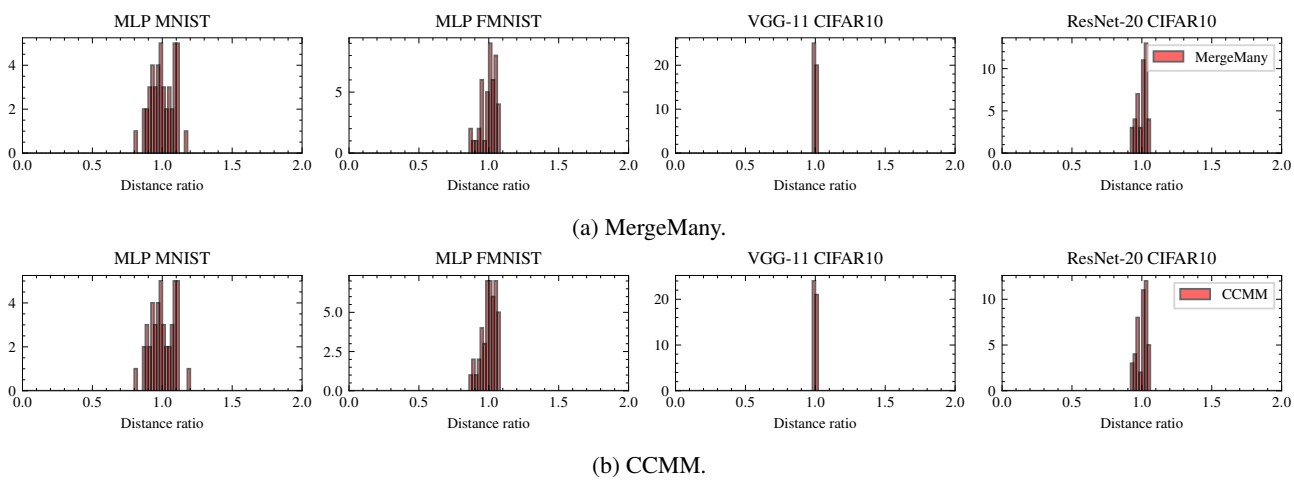

(a) MergeMany.

(b) CCMM.

Figure 9: Histogram of $L^2$ distances of all model pairs.

use a learning scheduler.

**VGG-11 and ResNet-20 on CIFAR10.** We utilized the VGG-11 and ResNet20 architectures of (Ainsworth et al., 2023). To accomplish linear mode connectivity (LMC), we increased the widths of VGG11 and ResNet20 by factors of 4 and 16, respectively. As described in (Jordan et al., 2023), we used the training dataset to repair the BatchNorm layers in these models during model merging. Optimization was conducted using SGD with a learning rate of $0.4$ and weight decay of $5 \times 10^{-4}$. The batch size and maximum number of epochs were set to 500 and 100, respectively. The following data augmentations were performed during training: random $32 \times 32$ pixel crops, and random horizontal flips. Based on the GitHub repository of (Jordan et al., 2023)[6], the learning scheduler was prepared according to the following code:

```python
# To execute this code, you need to import the packages as shown below in advance.
import numpy as np
import torch
# Here's the code for generating the learning rate scheduler.
# num_epochs: the number of epochs, opt: optimizer
lr_sch = np.interp(np.arange(1 + num_epochs), [0, 5, num_epochs], [0, 1, 0])
sch = torch.optim.lr_scheduler.LambdaLR(opt, lr_sch.__getitem__)
```

### B.2. Permutation Search

WM and STE were implemented following the GitHub repository of (Ainsworth et al., 2023)[7]. For STE and STE-MM, the learning rate, number of epochs, and batch size were set to 0.001, 10, and 256, respectively. As for CCMM, the source code from the authors' publicly available GitHub repository was used[8].

## C. Additional Experimental Results

### C.1. $L^2$ distance between all model pairs using MergeMany and CCMM

Figure 9 shows histograms of the $L^2$ distance ratios between all model pairs for MergeMany and CCMM. The figure indicates that the distances between model pairs are very close to their mean values, suggesting that they are nearly equivalent.

---

[6]https://github.com/KellerJordan/REPAIR
[7]https://github.com/samuela/git-re-basin
[8]https://github.com/crisostomi/cycle-consistent-model-merging

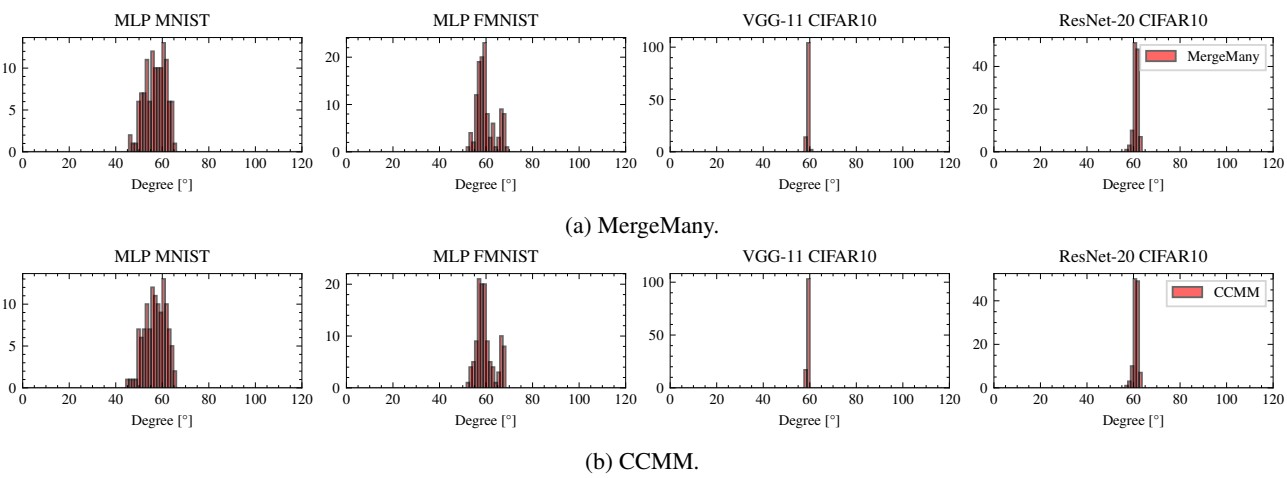

(a) MergeMany.

(b) CCMM.

Figure 10: Histograms of angles formed by all mode triplets.

## C.2. Angle of each triplet taken from the models

Figure 10 represents the histogram of the angles of all triplets from the models after applying the permutation. From the figure, most of the angles formed by the triplets are close to 60 degrees, suggesting, as described in Section 6.1, that the models after applying the permutation are close to a regular simplex.

## D. Proofs

### D.1. Finding $n-1$ permutations is sufficient

This subsection proves the following theorem:

**Theorem D.1.** *Let $g : \mathbb{R}^{d_{param}} \to \mathbb{R}$ be a function from the parameter space to the real numbers and invariant to permutations (i.e., for any parameter $\boldsymbol{\theta}$ and permutation $\pi$, $g(\pi(\boldsymbol{\theta})) = g(\boldsymbol{\theta})$ holds). Then, for any $n$ models $\boldsymbol{\theta}_1, \ldots, \boldsymbol{\theta}_n$, any permutations $\pi_1, \ldots, \pi_n$, and any scalars $s_1, \ldots, s_n \in \mathbb{R}$, there exist $n-1$ permutations $\pi'_1, \ldots, \pi'_{n-1}$ such that $g(\sum_{i \in [n]} s_i \pi_i(\boldsymbol{\theta}_i)) = g(\sum_{i \in [n-1]} s_i \pi'_i(\boldsymbol{\theta}_i) + s_n \boldsymbol{\theta}_n)$ holds. Here, $\pi'_1, \ldots, \pi'_{n-1}$ are independent of the function $g$.*

Note that the loss function $\mathcal{L}$ and $L^2$ norm are invariant to permutations.

*Proof.* We first prove the linearlity of permutaion $\pi = (\boldsymbol{P}_\ell)_{\ell \in [L]}$. For any two parameters $\boldsymbol{\theta} = \|_{\ell=1}^{L} (\text{vec}(\boldsymbol{W}_\ell) \| \boldsymbol{b}_\ell)$ and $\boldsymbol{\theta}' = \|_{\ell=1}^{L} (\text{vec}(\boldsymbol{W}'_\ell) \| \boldsymbol{b}'_\ell)$, and scalers $s, s' \in \mathbb{R}$, we consider $\pi(s\boldsymbol{\theta} + s'\boldsymbol{\theta}')$. From definition, since $s\boldsymbol{\theta} + s'\boldsymbol{\theta}' = \|_{\ell=1}^{L} ((\text{vec}(s\boldsymbol{W}_\ell + s'\boldsymbol{W}'_\ell)) \| (s\boldsymbol{b}_\ell + s'\boldsymbol{b}'_\ell))$ holds, we have

$$\pi(s\boldsymbol{\theta} + s'\boldsymbol{\theta}') = \|_{\ell=1}^{L} \left( (\text{vec}(s\boldsymbol{P}_\ell \boldsymbol{W}_\ell \boldsymbol{P}_{\ell-1}^\top + s'\boldsymbol{P}_\ell \boldsymbol{W}'_\ell \boldsymbol{P}_{\ell-1}^\top)) \| (s\boldsymbol{P}_\ell \boldsymbol{b}_\ell + s'\boldsymbol{P}_\ell \boldsymbol{b}'_\ell) \right) \tag{7}$$

$$= \|_{\ell=1}^{L} \left( s\boldsymbol{P}_\ell \boldsymbol{W}_\ell \boldsymbol{P}_{\ell-1}^\top \| s\boldsymbol{P}_\ell \boldsymbol{b}_\ell \right) + \|_{\ell=1}^{L} \left( s'\boldsymbol{P}_\ell \boldsymbol{W}'_\ell \boldsymbol{P}_{\ell-1}^\top \| s'\boldsymbol{P}_\ell \boldsymbol{b}'_\ell \right) \tag{8}$$

$$= s\|_{\ell=1}^{L} \left( \boldsymbol{P}_\ell \boldsymbol{W}_\ell \boldsymbol{P}_{\ell-1}^\top \| \boldsymbol{P}_\ell \boldsymbol{b}_\ell \right) + s'\|_{\ell=1}^{L} \left( \boldsymbol{P}_\ell \boldsymbol{W}'_\ell \boldsymbol{P}_{\ell-1}^\top \| \boldsymbol{P}_\ell \boldsymbol{b}'_\ell \right) \tag{9}$$

$$= s\pi(\boldsymbol{\theta}) + s'\pi(\boldsymbol{\theta}'), \tag{10}$$

which indicates that the permutation $\pi$ is a linear function.

Then, we prove that there exists permutations $\pi'_1, \pi'_2, \ldots, \pi'_{n-1}$ such that we have

$$g\left(\sum_{i \in [n]} s_i \pi_i(\boldsymbol{\theta}_i)\right) = g\left(\sum_{i \in [n-1]} s_i \pi'_i(\boldsymbol{\theta}_i) + s_n \boldsymbol{\theta}_n\right) \tag{11}$$

From the linearity of permutations and the invariance of $g$ with respect to permutations, we have

$$g\left(\sum_{i\in[n]} s_i \pi_i(\boldsymbol{\theta}_i)\right) = g\left(\pi_n \circ \pi_n^{-1}\left(\sum_{i\in[n]} s_i \pi_i(\boldsymbol{\theta}_i)\right)\right) = g\left(\pi_n\left(\sum_{i\in[n]} s_i \pi_n^{-1}\circ\pi_i(\boldsymbol{\theta}_i)\right)\right) = g\left(\sum_{i\in[n]} s_i \pi_n^{-1}\circ\pi_i(\boldsymbol{\theta}_i)\right).$$

$$\tag{12}$$

Thus, the permutations $\pi_1' = \pi_n^{-1}\circ\pi_1, \pi_2' = \pi_n^{-1}\circ\pi_2, \ldots, \pi_{n-1}' = \pi_n^{-1}\circ\pi_{n-1}$ satisfy Equation (11). In addition, the permutations $\pi_1', \ldots, \pi_n'$ are independent of the function $g$. $\qquad\square$

### D.2. Proof of Equation (6)

We prove the following theorem.

**Theorem D.2.** *Let $\boldsymbol{z}$ be a Rademacher random vector, $\boldsymbol{\theta}$ be parameters of an NN model, and $\boldsymbol{H}$ be the Hessian matrix of the loss function at the model $\boldsymbol{\theta}$. Then, we have*

$$\mathrm{tr}(\boldsymbol{H}\odot\boldsymbol{\theta}\boldsymbol{\theta}^\top) = \mathbb{E}_{\boldsymbol{z}}(\boldsymbol{z}\,\mathrm{diag}(\boldsymbol{\theta}))^\top \boldsymbol{H}(\boldsymbol{z}\,\mathrm{diag}(\boldsymbol{\theta})). \tag{13}$$

*Proof.* With the Huchinson trace estimator (Hutchinson, 1990), we have

$$\mathrm{tr}(\boldsymbol{H}\odot(\boldsymbol{\theta}\boldsymbol{\theta}^\top)) = \mathrm{tr}(\boldsymbol{H}\,\mathrm{diag}(\boldsymbol{\theta})\,\mathrm{diag}(\boldsymbol{\theta})) = \mathrm{tr}(\mathrm{diag}(\boldsymbol{\theta})\boldsymbol{H}\,\mathrm{diag}(\boldsymbol{\theta})) \tag{14}$$

$$= \mathbb{E}_{\boldsymbol{z}}\boldsymbol{z}^\top\,\mathrm{diag}(\boldsymbol{\theta})\boldsymbol{H}\,\mathrm{diag}(\boldsymbol{\theta})\boldsymbol{z} = \mathbb{E}_{\boldsymbol{z}}(\mathrm{diag}(\boldsymbol{\theta})\boldsymbol{z})^\top\boldsymbol{H}(\boldsymbol{z}\,\mathrm{diag}(\boldsymbol{\theta})). \tag{15}$$

$$\square$$

