# OpenReview forum: "Linear Mode Connectivity between Multiple Models modulo Permutation Symmetries"
_ICML.cc/2025/Conference — ICML 2025 poster_

### Official Review · Reviewer_K8ki · 2025-03-06

**Overall Recommendation:** 3

**Summary:**

The authors observe the linear mode connectivity hypothesis as proposed before has only been confirmed between two independent models, and propose an algorithm to merge multiple models such that the test loss doesn’t meaningfully change but the model has an approximately flat global minima.

**Claims And Evidence:**

The claims seem to be well-supported empirically, that the merging of two models doesn't scale and this new method does.

**Essential References Not Discussed:**

N/A

**Experimental Designs Or Analyses:**

Yes

**Methods And Evaluation Criteria:**

Yes

**Other Comments Or Suggestions:**

N/A

**Other Strengths And Weaknesses:**

N/A

**Questions For Authors:**

N/A

**Relation To Broader Scientific Literature:**

The contribution seems important for the larger literature on linear mode connectivity, but this isn't particularly well motivated in the paper itself.  The STE-MM algorithm seems novel, but because the methods either lower test accuracy or provide a very small increase, my understanding is that these algorithms are not intended to improve metrics for their own sake but to give more evidence that linear mode connectivity of independently trained networks is empirically true.  I think this requires a bit more discussion in the paper.  In particular, the fact that the convex basin of merging multiple models becomes less sharp is intuitive, but the consequences of this fact aren’t elaborated on much.

**Theoretical Claims:**

Yes

---

> ### Author Rebuttal · Authors · 2025-03-28
>
> Thank you very much for your thoughtful comments and for taking the time to carefully read our paper.
>
> > The contribution seems important for the larger literature on linear mode connectivity, but this isn't particularly well motivated in the paper itself. The STE-MM algorithm seems novel, but because the methods either lower test accuracy or provide a very small increase, my understanding is that these algorithms are not intended to improve metrics for their own sake but to give more evidence that linear mode connectivity of independently trained networks is empirically true. I think this requires a bit more discussion in the paper. In particular, the fact that the convex basin of merging multiple models becomes less sharp is intuitive, but the consequences of this fact aren’t elaborated on much.
>
> Thank you for your thoughtful feedback. As you pointed out, we believe that our work contributes to the broader understanding of linear mode connectivity (LMC), particularly in the context of merging multiple independently trained models. While we touch on this connection in Section 1 of the paper, it may not have been sufficiently emphasized. To better reflect this perspective, we are considering revising the introduction to explicitly highlight the relevance to LMC and potentially modifying the title to more directly reflect this connection, for example, “Linear Mode Connectivity for Multiple Models Modulo Permutation Symmetries.”
>
> We use average-case sharpness as a metric because, as discussed in Andriushchenko et al. (2023), it quantifies how much the loss increases under Gaussian perturbations of model parameters. If multiple models can be transferred via permutations into the same convex basin, we expect the loss around their midpoint to remain relatively flat—reflected in low average-case sharpness.
>
> Indeed, our experiments show that STE-MM consistently reduces average-case sharpness around merged models. While this does not provide direct evidence of a single convex basin, it supports the possibility that such a basin exists. A more fine-grained analysis of this phenomenon remains an important direction for future research, which we intend to pursue.

---

> > ### Comment · Reviewer_K8ki · 2025-04-01
> >
> > I appreciate the author's response.  I will keep my score.

---

### Official Review · Reviewer_SJMK · 2025-03-11

**Overall Recommendation:** 4

**Summary:**

Prior work showed that linear mode connectivity can be achieved between two independently trained neural networks by applying an appropriate parameter permutation, suggesting that SGD-trained models converge to a shared low-loss basin under permutation symmetries. This paper extends their analysis to multiple models, proposing STE-MM, a runtime-optimized permutation search method that maintains low test loss after merging and reduces loss sharpness as the number of models increases, indicating that linear connectivity generalizes to multiple models.

## Update after rebuttal
I thank the authors for the rebuttal. I maintain my score.

**Claims And Evidence:**

Yes, I find the results in the paper highly interesting and the work exceptionally well executed. I especially appreciate the approach of optimizing the permutation matrix search by reusing dual variables from previous iterations to dynamically adjust the cost matrix, significantly improving the efficiency of the LAP solver.

**Essential References Not Discussed:**

The references are appropriate.

**Experimental Designs Or Analyses:**

The presented results are compelling and should be well-received by the conference community.

**Methods And Evaluation Criteria:**

Yes, the proposed approach and insights demonstrate a strong understanding of classical algorithms. The choice of datasets and baselines is appropriate. I recommend that the authors extend their approach to more challenging datasets, such as CIFAR-100 and ImageNet, to further validate its effectiveness.

**Other Comments Or Suggestions:**

- Typo "mdoel" on line 060, second column
- Please revise lines 115-134 in the second column to ensure that the referenced conjecture is clearly presented as a conjecture rather than a theorem.
- Line 350: "While the loss of the merged model also tends to in crease with the number of models in STE-MM, the amount of increase in test loss value becomes small as the number of models grows." - I think it's hard to see this in the provided figures, but could be shown in a dedicated plot.
- I suggest to move Fig. 7 and Fig. 8 to the main paper - these are quite essential.

**Other Strengths And Weaknesses:**

The paper presents a valuable result that advances the theoretical understanding of deep learning and introduces a method that is highly relevant for applications such as multi-task and distributed learning. I don't see major weaknesses.

**Questions For Authors:**

- Since STE-MM is the first method to empirically place multiple models into a single basin, could it be used to analyze potential biases or inefficiencies in other, more lightweight, permutation search techniques?

- Could you comment on how your paper relates to the concurrent work https://arxiv.org/abs/2403.07968?

**Relation To Broader Scientific Literature:**

Both the proposed optimization method (STE-MM) and the findings presented in the paper make a significant contribution to the scientific literature on validating the permutation hypothesis.

**Theoretical Claims:**

The theoretical and algorithmic aspects of the paper are well-explained and appear solid.

---

> ### Author Rebuttal · Authors · 2025-03-28
>
> We are grateful for your valuable comments and for your close reading of the paper.
>
> > I recommend that the authors extend their approach to more challenging datasets, such as CIFAR-100 and ImageNet, to further validate its effectiveness.
>
> Thank you for your suggestion. To test our method on more complex tasks, we performed an experiment on model merging using ResNet-50 models trained on the ImageNet dataset. The following table presents the test losses of merged models using various permutation search methods, including STE-MM. Results are shown for up to six models due to GPU memory limitations. The ResNet-50 models were trained using a training script published on GitHub by the FFCV library (i.e., https://github.com/libffcv/ffcv-imagenet). The table reports the average and standard deviation of the test loss across three model merging trials. As seen in the table, the test loss of the merged model using STE-MM decreases monotonically as the number of merged models increases, while that of other permutation search methods increases, demonstrating the effectiveness of our method. Note that the test loss of the merged model could potentially be improved by increasing the model width, even when multiple models are merged. Ainsworth et al. (2023), for instance, have shown that the test accuracy of the merged model improves with increased model width when two ResNet-50 models are merged.
>
> **Test loss of the merged model when ResNet-50 models trained on ImageNet are combined**
>
> | #models | MergeMany         | CCMM              | STE-MM            |
> |---:|:------------------|:------------------|:------------------|
> |  2 | $5.835 \pm 0.134$ | $5.822 \pm 0.066$ | $5.306 \pm 0.022$ |
> |  3 | $6.341 \pm 0.023$ | $6.399 \pm 0.062$ | $5.179 \pm 0.044$ |
> |  4 | $6.555 \pm 0.034$ | $6.576 \pm 0.047$ | $4.887 \pm 0.039$ |
> |  5 | $6.661 \pm 0.084$ | $6.657 \pm 0.040$ | $4.684 \pm 0.010$ |
> |  6 | $6.689 \pm 0.047$ | $6.677 \pm 0.037$ | $4.559 \pm 0.027$ |
>
> > Typo "mdoel" on line 060, second column
> > Please revise lines 115-134
> > could be shown in a dedicated plot.
>
> We apologize for any typos, unclear figures, or ambiguous expressions in the text. We will carefully revise the manuscript to address all of these issues in the camera-ready version.
>
> > I suggest to move Fig. 7 and Fig. 8 to the main paper.
>
> Thank you for your suggestion. As you mentioned, these are important results, so we will include Figures 7 and 8 in the main body of the paper.
>
> > could it be used to analyze potential biases or inefficiencies in other, more lightweight, permutation search techniques?
>
> This is a valuable perspective. Since methods based on $L^2$ distance, such as MergeMany and CCMM, do not require training, they can search for permutation matrices with low computational cost. However, our experimental results show that the performance of the merged model deteriorates as the number of models being merged increases when using these methods. This suggests a fundamental difference between the permutation matrices discovered by STE-MM and those identified by lightweight approaches. A more in-depth comparison of these permutation matrices may help uncover the causes of inefficiency in the latter and guide improvements in their design.
>
> > Could you comment on how your paper relates to the concurrent work https://arxiv.org/abs/2403.07968?
>
> Thank you for the reference. We were not previously aware of this paper. It proposes the star domain conjecture and introduces the Starlight algorithm to verify it by identifying a parameter $\theta^\ast$ from multiple SGD solutions. The authors show empirically that such a $\theta^\ast$ exists even for models with smaller widths. Entezari et al.'s conjecture (i.e., multiple SGD solutions can be transferred into a single approximately convex basin via permutations) makes a stronger claim than the star domain conjecture, as shown in the figure on page 2 of their paper.
>
> This result is particularly interesting, as the method performs well in the small-width regime. In contrast, our STE-MM approach requires sufficiently wide models to transfer them into a shared approximately convex basin; otherwise, a significant barrier remains.
>
> It is unclear whether this limitation arises from the difficulty of searching the large permutation space—due to the discrete nature of the optimization—or from a more fundamental property of the loss landscape.
>
> If the latter is the case, it may suggest that wider (i.e., overparameterized) models tend to induce simpler loss landscapes. In fact, a recent theoretical study (https://openreview.net/forum?id=4xWQS2z77v) on two-layer neural networks has shown that increasing the width leads to a simpler structure in the loss landscape. It is plausible that a similar phenomenon occurs in deeper networks as well. We believe that further investigation into this aspect is a promising direction for future research.

---

### Official Review · Reviewer_sFBh · 2025-03-12

**Overall Recommendation:** 3

**Summary:**

This paper investigates permutation-based methods for merging multiple models. The literature mainly focused on merging pairs of models, and this paper shows that these methods fail to transfer multiple models into the same basin. Then they introduce a method for merging multiple methods and find multiple permutations extending the Stringh Through Estimator method and propose an accelerated version of weight matching for faster permutation search

**Claims And Evidence:**

Claims are supported by the experiments.

**Essential References Not Discussed:**

NA

**Experimental Designs Or Analyses:**

The experimental design is sound, all claims are supported by experiments.

**Methods And Evaluation Criteria:**

The paper is clear and introduces experiments to motivate the necessity of a new method to tackle the problem of merging multiple models. Then the proposed method is benchmarked against the correct set of baselines and it scales much better with the number of merged models in terms of loss.

**Other Comments Or Suggestions:**

typos:
190: Permuta*ion Search for Multiple Models
403 right - experime*tnal results

**Other Strengths And Weaknesses:**

Strengths:
- important problem
- scalable method over the number of models
- experiments are sound and show improvement over the baselines
- interesting analysis on the flatness of the final solution

Weaknesses:
- The algorithm explanation could be improved. I suggest the authors expand and clarify the reason for dummy variables and how the algorithm works in general by following the explanation of the seminal paper (Ainsworth et al 2023), section 3.3.

- Experiments are only performed on academic benchmarks such as CIFAR10 and MNIST. I’m sorry about this comment, I hope it does not trivialize your analysis, but I recommend testing the scalability of the method on more complex tasks such as ImageNet.

**Questions For Authors:**

please see weaknesses and strengths, and:

- can the authors provide an analysis of the permutation statistics, for example, what is the percentage of the weights that are being permuted?
- another thing is, can the authors provide a baseline where no permutation is applied? I think it is important to establish that there are actual permutations in the considered setting.

**Relation To Broader Scientific Literature:**

yes. It connects to seminal papers on linear mode connectivity, permutation-based methods and also recent works on merging multiple models (Crisostomi et al 2024)

**Theoretical Claims:**

There is a theoretical section (proof in appendix) claiming that finding n-1 permutations is sufficient. This result serves as a base for their algorithm 1. The proof seems correct.

---

> ### Author Rebuttal · Authors · 2025-03-28
>
> We appreciate your insightful feedback and the effort you put into reviewing our work.
>
> > The algorithm explanation could be improved. I suggest the authors expand and clarify the reason for dummy variables and how the algorithm works in general by following the explanation of the seminal paper (Ainsworth et al 2023), section 3.3.
>
> Thank you for your suggestion on improving the explanation. We will revise the description of STE-MM to make it clearer, drawing on the explanation provided in Ainsworth et al. (2023), Section 3.3.
>
> > Experiments are only performed on academic benchmarks such as CIFAR10 and MNIST.
> I’m sorry about this comment, I hope it does not trivialize your analysis, but I recommend testing the scalability of the method on more complex tasks such as ImageNet.
>
> Thank you for your suggestion. To evaluate the scalability of our method, we conducted additional experiments of model merging with ResNet-50 models trained on the ImageNet dataset. Due to space limitations in the rebuttal, the detailed experimental results are included in our response to Reviewer SJMK’s first comment. The results demonstrate that our method scales effectively to more complex tasks such as ImageNet.
>
>
> > typos: 190: Permutaion Search for Multiple Models 403 right - experimetnal results
>
> We are sorry for the typos. We will carefully review the entire paper and make the necessary corrections.
>
> > can the authors provide an analysis of the permutation statistics, for example, what is the percentage of the weights that are being permuted?
>
> Yes, certainly. The following table provides an analysis of how closely the permutation matrices discovered by STE-MM resemble the identity matrix. Specifically, we measured this by counting the number of entries equal to 1 in the diagonal of each permutation matrix and dividing this count by the size of the matrix. This matching rate was calculated for each layer, then averaged across all layers in a model, and finally averaged across all models being merged. The table reports the mean and standard deviation over three model merging trials. As shown, the matching ratio remains close to zero regardless of the number of models, indicating that the found permutation matrices are significantly different from the identity matrix.
>
> **Percentage of matches between the permutation matrix and the identity matrix [\%]**
>
> | #models | MLP, MNIST        | MLP, FMNIST       | VGG-11, CIFAR10   | ResNet-20, CIFAR10   |
> |---:|:------------------|:------------------|:------------------|:---------------------|
> |  2 | $0.174 \pm 0.099$ | $0.217 \pm 0.136$ | $0.336 \pm 0.122$ | $0.217 \pm 0.033$    |
> |  3 | $0.141 \pm 0.082$ | $0.228 \pm 0.117$ | $0.298 \pm 0.258$ | $0.191 \pm 0.051$    |
> |  4 | $0.152 \pm 0.075$ | $0.217 \pm 0.075$ | $0.254 \pm 0.126$ | $0.242 \pm 0.023$    |
> |  5 | $0.157 \pm 0.052$ | $0.212 \pm 0.033$ | $0.171 \pm 0.012$ | $0.203 \pm 0.002$    |
> |  6 | $0.208 \pm 0.069$ | $0.152 \pm 0.015$ | $0.174 \pm 0.036$ | $0.227 \pm 0.012$    |
> |  7 | $0.177 \pm 0.023$ | $0.181 \pm 0.070$ | $0.138 \pm 0.043$ | $0.221 \pm 0.027$    |
> |  8 | $0.158 \pm 0.052$ | $0.180 \pm 0.019$ | $0.160 \pm 0.034$ | $0.193 \pm 0.006$    |
> |  9 | $0.146 \pm 0.008$ | $0.217 \pm 0.012$ | $0.141 \pm 0.032$ | $0.227 \pm 0.046$    |
> | 10 | $0.178 \pm 0.044$ | $0.219 \pm 0.018$ | $0.176 \pm 0.068$ | $0.232 \pm 0.010$    |
>
> > another thing is, can the authors provide a baseline where no permutation is applied? I think it is important to establish that there are actual permutations in the considered setting.
>
> The following table shows the test accuracy of the merged model when the models are combined without applying any permutation matrices. As shown, the accuracy drops significantly as the number of models being merged increases. This result highlights the importance of searching for appropriate permutation matrices when merging multiple models.
>
> **Test accuracy of the merged model without permutation**
>
> | #models | MLP, MNIST         | MLP, FMNIST        | VGG-11, CIFAR10    | ResNet-20, CIFAR10   |
> |---:|:-------------------|:-------------------|:-------------------|:---------------------|
> |  2 | $82.223 \pm 5.318$ | $46.930 \pm 7.659$ | $80.353 \pm 0.365$ | $86.653 \pm 1.317$   |
> |  3 | $23.020 \pm 9.077$ | $14.340 \pm 2.318$ | $32.777 \pm 4.380$ | $45.780 \pm 8.533$   |
> |  4 | $9.760 \pm 0.017$  | $10.043 \pm 0.075$ | $10.000 \pm 0.000$ | $10.267 \pm 0.281$   |
> |  5 | $9.740 \pm 0.000$  | $10.003 \pm 0.006$ | $10.000 \pm 0.000$ | $10.013 \pm 0.023$   |
> |  6 | $9.740 \pm 0.000$  | $10.010 \pm 0.017$ | $10.000 \pm 0.000$ | $10.000 \pm 0.000$   |
> |  7 | $9.740 \pm 0.000$  | $10.000 \pm 0.000$ | $10.000 \pm 0.000$ | $10.000 \pm 0.000$   |
> |  8 | $9.740 \pm 0.000$  | $10.000 \pm 0.000$ | $10.000 \pm 0.000$ | $10.000 \pm 0.000$   |
> |  9 | $9.740 \pm 0.000$  | $10.000 \pm 0.000$ | $10.000 \pm 0.000$ | $10.000 \pm 0.000$   |
> | 10 | $9.740 \pm 0.000$  | $10.000 \pm 0.000$ | $10.000 \pm 0.000$ | $10.000 \pm 0.000$   |

---

### Official Review · Reviewer_AEKC · 2025-03-13

**Overall Recommendation:** 3

**Summary:**

This paper focuses on the linear mode connectivity between neural networks (NNs) trained using stochastic gradient descent (SGD). First, it shows that existing permutation search methods perform poorly when more than two models are involved. To address this issue, the authors propose a novel search method, the Straight-Through Estimator for Multiple Models (STE-MM). Empirical results show that this method is both effective and efficient in improving linear mode connectivity among multiple models.

**Claims And Evidence:**

The authors claim that previous methods fail to achieve linear mode connectivity among more than three models. This is demonstrated in the experiments in Section 3, where, even when the remaining models are transformed into one specific model—thus theoretically entering the loss basin of that model—merging these models still leads to an increase in loss and a decrease in accuracy.

The authors propose a new search method, STE-MM, to efficiently transfer multiple models into a single loss basin using a permutation matrix. They conduct experiments on MLP, VGG-11, and ResNet-20 trained on MNIST, FMNIST, and CIFAR-10, comparing their method with MergeMany and CCMM. The results indicate that: (1) the acceleration method introduced in Section 4.2 speeds up the search process; (2) their method generally outperforms others in terms of loss evaluation and accuracy on benchmark datasets; (3) they define the sharpness of the loss at the center point and demonstrate that it decreases.

**Essential References Not Discussed:**

Most essential related works, as I know, are mentioned.

**Experimental Designs Or Analyses:**

See Claims and Evidence.

**Methods And Evaluation Criteria:**

The models and evaluation criteria are well-suited for the field of computer vision. Although they do not involve NLP, this is not a limitation in the given context.

**Other Comments Or Suggestions:**

- The clarity of Figure 5 could be improved. It appears to illustrate that the distance ratio of STE-MM is more concentrated; however, the overlapping histograms make it difficult to discern specific details. Enhancing the visualization, such as by adjusting transparency or using distinct color schemes, may help improve readability.
- Typos: 059 (right column) model; 061 (right column) denotes.

**Other Strengths And Weaknesses:**

S1. The paper writing is quite well and clear.

S2. See **Claims and Evidence.**

W1. See **Relation To Broader Scientific Literature.**

**Questions For Authors:**

- Could STE-MM also assist in determining the optimal hyperparameters for different models? For instance, in model merging, where different models have distinct weights, can STE-MM help in assigning appropriate values?

**Relation To Broader Scientific Literature:**

The paper proposes a merging method grounded in theoretical principles, which has been empirically demonstrated to be effective for model merging involving multiple models. However, other merging methods, such as TIES merging and DARE, are also widely used and have been shown to be effective. Additionally, due to computational cost considerations, simple averaging remains a common approach in multiple-model merging.

**Theoretical Claims:**

The theoretical claims see the (4) in Supplementary materials.

---

> ### Author Rebuttal · Authors · 2025-03-28
>
> Thank you for reading our paper carefully and for your constructive comments.
>
> > However, other merging methods, such as TIES merging and DARE, are also widely used and have been shown to be effective. Additionally, due to computational cost considerations, simple averaging remains a common approach in multiple-model merging.
>
> Thank you for your comment. While those methods are indeed important prior works, their assumptions differ significantly from ours. Specifically, TIES and DARE propose methods for merging multiple fine-tuned models based on a shared pre-trained model. In contrast, we focus on merging models that have been trained from scratch using different random seeds. Previous studies, such as model soups (Wortsman et al. 2022), have shown that simply averaging the weights of fine-tuned models originating from a common base model can improve performance. However, when merging models trained from scratch with different seeds, it has been observed that naive weight averaging can substantially degrade performance. Our method addresses the more challenging scenario of merging such models effectively, which we consider one of our key contributions.
>
> > The clarity of Figure 5 could be improved.
>
> We apologize for Figure 5's poor visibility. As you pointed out, we will revise it in the camera-ready version to improve its readability.
>
> > Typos: 059 (right column) model; 061 (right column) denotes.
>
> Thank you for pointing out the typos. We will thoroughly proofread the entire manuscript to ensure that all typos are corrected.
>
> > Could STE-MM also assist in determining the optimal hyperparameters for different models? For instance, in model merging, where different models have distinct weights, can STE-MM help in assigning appropriate values?
>
> Thank you for your question. If we understand correctly, you may be referring to whether STE-MM could be extended to assist with hyperparameter tuning when merging models with distinct weights. In its current form, STE-MM does not explicitly address hyperparameter optimization—its objective is to discover effective permutations for merging models.
> The algorithm itself only includes standard training hyperparameters such as learning rate and batch size. For example, in Equation 5, the values of $\lambda_1, \ldots, \lambda_n$ are drawn from a uniform distribution and are not considered hyperparameters. Therefore, applying STE-MM to hyperparameter selection would require substantial modification, as this lies outside its intended scope.

---

> > ### Comment · Reviewer_AEKC · 2025-04-03
> >
> > Thank you for the helpful clarifications. The focus of merging models that have been trained from scratch addresses a more challenging issue. This will not affect my score for the paper, but i would like to clarify: is the aim of merging to achieve better generalization?
> >
> > Additionally, if possible, I would still appreciate it if the authors could improve the presentation of Figure 5 to enhance the overall appearance of the paper.

---

> > > ### Author Response · Authors · 2025-04-04
> > >
> > > Thank you for your additional questions.
> > >
> > > > This will not affect my score for the paper, but i would like to clarify: is the aim of merging to achieve better generalization?
> > >
> > > Obtaining a more generalized model is not the primary objective of this paper; rather, we consider it one of the potential outcomes resulting from our main goal. The objective of our study is to investigate whether Linear Mode Connectivity (LMC) holds across multiple models. In other words, we aim to determine whether it is possible to find suitable permutations such that multiple models can be transferred into the same approximately convex basin.
> > >
> > > As described in the introduction of the paper, this inquiry is driven by a scientific interest in understanding why Stochastic Gradient Descent (SGD) is so effective in training neural networks, as discussed in works such as Entezari et al. and Ainsworth et al. Therefore, the goal of our work is rooted more in scientific curiosity than in practical utility.
> > >
> > > That said, if we are able to find permutations that enable LMC to hold among multiple models, the resulting merged model could potentially outperform the original models. In fact, as shown in Figure 2(a), the test accuracy of the merged model improves as the number of models being merged increases. Furthermore, as shown in Figure 6, the loss landscape around the merged model becomes flatter. These observations suggest that model merging may lead to improved generalization. In this sense, although enhancing generalization is not the direct objective of our work, we believe it is a possible byproduct of the proposed model merging approach.
> > >
> > > > if possible, I would still appreciate it if the authors could improve the presentation of Figure 5 to enhance the overall appearance of the paper.
> > >
> > > We have uploaded the revised figures at the following URL—please have a look:
> > > https://anonymous.4open.science/r/ICML_rebuttal_figs-3029/figures.pdf
> > >
> > > In response to the concern that the differences between methods were hard to distinguish, we have now plotted the results separately for STE-MM, MergeMany, and CCMM. The top three figures in the PDF show histograms of the $L^2$ distances for all model pairs, while the bottom three show histograms of the angles formed by all model triplets.
> > >
> > > We apologize for not having conveyed this clearly in our previous response, but our intention with Figure 5 was to demonstrate that, regardless of the permutation search method used, the distances for model pairs after permutation tend to be roughly equivalent. The point was not necessarily to highlight that STE-MM results are more tightly concentrated around 1.0.
> > >
> > > As seen in the updated figures, similar trends are observed across all methods. Therefore, we plan to replace Figure 5 in the main text with the top figure from the PDF (i.e., the histogram of model pair distances after permutation using STE-MM). The remaining figures will be added to the appendix. We will also revise the main text to note that results from other methods are available in the appendix and that they exhibit similar behavior.

---

### Decision · Program_Chairs · 2025-05-01

**Decision:**

Accept (poster)

**Comment:**

The paper addresses the challenge of achieving linear mode connectivity (LMC) among multiple neural networks (NNs) trained using SGD. The prior works experience degradation in performance when merging > 2 models. The current work introduces a novel method, Straight-Through Estimator for Multiple Models (STE-MM), that directly optimizes for loss barrier. The authors also provide a method to accelerate the permutation search. The proposed method outperforms existing approaches in improving LMC, demonstrated through experiments on multiple datasets and network architectures.

Overall, the reviewers are in consensus that \
(a) the paper tackles a relevant problem and addresses the limitation of prior work in LMC\
(b) the proposed method STE-MM is novel and constitutes a significant contribution. The reviewers appreciate the optimization approach. \
(c) The empirical evaluations demonstrate the effectiveness of the method. Experiment design is sound. Authors also provide results from ImageNet in rebuttal. \
(d) Paper is well written.

I would recommend a solid Accept with following improvements as pointed by the reviewers \
(1) Improving clarity in Fig 5 by revision\
(2) Proofread and correct typos as pointed by reviewers\
(3) Please revise the title to highlight relevant to LMC\
(4) If possible, moving Fig 7 & 8 to main body